# The Elevated De Ritis Ratio on Admission Is Independently Associated with Mortality in COVID-19 Patients

**DOI:** 10.3390/v14112360

**Published:** 2022-10-26

**Authors:** Bálint Drácz, Diána Czompa, Katalin Müllner, Krisztina Hagymási, Pál Miheller, Hajnal Székely, Veronika Papp, Miklós Horváth, István Hritz, Attila Szijártó, Klára Werling

**Affiliations:** Department of Surgery, Transplantation and Gastroenterology, Semmelweis University, Üllői út 26, 1085 Budapest, Hungary

**Keywords:** COVID-19, mortality, De Ritis ratio, AST, ALT, IL-6, Child–Pugh score, liver injury

## Abstract

Liver damage in COVID-19 patients was documented as increased alanine aminotransferase (ALT) and aspartate aminotransferase (AST) levels or an elevated AST/ALT ratio, known as the De Ritis ratio. However, the prognostic value of the elevated De Ritis ratio in COVID-19 patients is still unknown. The aim of our study was to evaluate the prognostic value of the De Ritis ratio compared to other abnormal laboratory parameters and its relation to mortality. We selected 322 COVID-19 patients in this retrospective study conducted between November 2020 and March 2021. The laboratory parameters were measured on admission and followed till patient discharge or death. Of the 322 COVID-19 patients, 57 (17.7%) had gastrointestinal symptoms on admission. The multivariate analysis showed that the De Ritis ratio was an independent risk factor for mortality, with an OR of 29.967 (95% CI 5.266–170.514). In ROC analysis, the AUC value of the the De Ritis ratio was 0.85 (95% CI 0.777–0.923, *p* < 0.05) with sensitivity and specificity of 80.6% and 75.2%, respectively. A De Ritis ratio ≥1.218 was significantly associated with patient mortality, disease severity, higher AST and IL-6 levels, and a lower ALT level. An elevated De Ritis ratio on admission is independently associated with mortality in COVID-19 patients, indicating liver injury and cytokine release syndrome.

## 1. Introduction

The severe acute respiratory syndrome coronavirus 2 causing Coronavirus disease 2019 (COVID-19) was first isolated in humans in Wuhan, China, in December 2019. The virus has since spread over the world, affecting more than 200 countries and regions [1]. The WHO declared the COVID-19 outbreak a pandemic on 11 March 2020 [2]. As of February, 2022 there have been approximately 376 million confirmed cases and more than 5.5 million deaths worldwide [3]. In Hungary, there have been more than 1.5 million confirmed cases of COVID-19 with more than 41,000 deaths reported to the WHO [4]. Previous studies showed that up to 26% of patients with COVID-19 developed gastrointestinal symptoms dominated by diarrhea, nausea, and loss of appetite [5,6]. In 14–53% of COVID-19 patients, the disease was associated with hepatic dysfunction or liver damage, as indicated by elevated transaminases (AST, ALT) and cholestatic parameters (GGT, total bilirubin, ALP), and hepatic involvement was linked to a more severe outcome [7]. The pathophysiology of liver injury in COVID-19 is multifactorial. The virus may bind to ACE2-positive cholangiocytes and cause a direct cytopathic effect by impairing the barrier and bile acid transport functions of cholangiocytes. Dysregulation of the innate immune system may trigger lymphopenia and abnormal inflammatory cytokine levels (IL-2, IL-6, IL-10) and thereby contribute to the onset of a potentially life-threatening cytokine storm. Hypoxic conditions due to sinusoidal microembolisations and acute respiratory distress syndrome (ARDS) may induce a congestive hepatopathy. Significantly increased transaminase levels indicative of drug-induced liver injury (DILI) have already been documented with the use of tocilizumab [5,8,9,10].

Generally, in 2–11% of COVID-19 cases, an underlying liver disease was recorded in the medical history. The clinical outcome of COVID-19 patients may be influenced by the cause of liver disease. Liver cirrhosis is a high risk factor for COVID-19 mortality. Among patients with cirrhosis as measured by the Child–Pugh (CP) score, a strong association was found between the severity of cirrhosis and mortality [11]. Cirrhosis-associated immune dysfunction (CAID) is a common condition in patients with chronic liver disease, in particular with liver cirrhosis. Upregulation of macrophages and the complement system as well as impaired lymphocytes and neutrophils, along with intestinal dysbiosis, may lead to an aberrant inflammatory response during infection [12,13]. Therefore, CAID makes COVID-19-infected individuals more susceptible to bacterial and fungal infections [14,15]. Moreover, COVID-19 infection may lead to acute-on-chronic liver failure (ACLF), a syndrome characterized by an acute decompensation of chronic liver disease associated with multiple organ failure (MOF) and increased mortality rate [16].

## 2. Materials and Methods

### 2.1. Study Population

In the third wave of the COVID-19 pandemic, between November 2020 and March 2021, 322 COVID-19 patients were retrospectively recruited at the Department of Surgery, Transplantation, and Gastroenterology, Semmelweis University.

COVID-19 was confirmed by a positive reverse-transcription–polimerase chain reaction (RT-PCR, SEQONCE qPCR Multi Kit, IVD) test using the protocol by the World Health Organization [17]. Oropharyngeal or/and nasopharyngeal swab specimens were collected, and high-resolution computer tomography (HRCT, Phillips Incisive128) was performed on admission and during the hospital stay.

### 2.2. Study Design

This analysis of the retrospective study was conducted using data from electronic medical records. All data of confirmed COVID-19 patients regarding epidemiological and clinical characteristics, laboratory findings, imaging features, management, and treatment were collected and reviewed. The study was approved by the Scientific and Research Ethics Committee of the Medical Research Council of Hungary (IV/5245-1/2021/EKU, Budapest, 6 July 2021) and conforms to the ethical norms and standards laid down in the Declaration of Helsinki.

On admission, a physical examination was performed, and a detailed, accurate medical history was recorded with special emphasis on comorbidities, symptoms (respiratory tract and gastrointestinal manifestations), and underlying liver diseases. We also collected data about pulmonary imaging features and intensive care received, including the use of any respiratory support. Hence, patients confirmed with COVID-19 were divided into mild, moderate, severe, and critical classes with regard to the clinical course. The clinical classification and diagnostic criteria of COVID-19 were based on our national protocol published by the Ministry of Human Resources according to the guidelines of the Chinese National Health Commission [18].

### 2.3. Data Collection

The time of hospital admission was defined as the number of days from the onset of symptoms to hospital admission. Laboratory biomarkers were systematically collected in the acute phase of COVID-19 infection at hospital admission. During hospitalization, follow-up laboratory examinations were performed regularly, including the measurement of liver enzymes (AST, ALT), cholestatic parameters (GGT, total bilirubin, ALP), inflammatory biomarkers (baseline C-reactive protein, IL-6, PCT), and a liver function marker (albumin). Laboratory markers such as direct bilirubin, LDH, blood cell counts (erythrocytes, leukocytes, thrombocytes), and INR were excluded from this study because laboratory tests of these markers were not carried out on all patients with COVID-19. Furthermore, at each follow-up, the De Ritis ratio (AST/ALT) was calculated. Patient surveillance was maintained till discharge or death. The criteria for releasing COVID-19 patients from our hospital were as follows: (1) resolution of fever for >48 h without antipyretics, (2) oxygen saturation ≥ 94%, (3) no signs of increased work of breathing or respiratory distress, (4) improvement in the signs and symptoms of illness (cough, shortness of breath, and oxygen requirement), and (5) two negative RT-PCR tests in a row, at least 24 h apart.

Admitted COVID-19 patients with preexisting liver disease were divided into two groups: patients having chronic liver disease (CLD) with cirrhosis and those having CLD without cirrhosis. Data about the etiology of CLD in both groups were retrieved from individual clinical reports. In addition, we used the Child–Pugh (CP) score, based on our local protocol, for the assessment of prognosis in patients with liver cirrhosis. The modified Child–Pugh classification included the serum concentrations of total bilirubin and albumin, the international normalized ratio (INR), the degree of ascites, and the degree of hepatic encephalopathy. A CP score of 5 to 6 was considered as CP class A (well-compensated disease), of 7 to 9 as CP class B (significant functional compromise), and of 10 to 15 as CP class C (decompensated cirrhosis) [19].

### 2.4. Statistical Analysis

All statistical analyses were performed using SPSS software version 28 (IBM Corporation Armonk, NY, USA). The data were tested for normality using the Kolmogorov–Smirnov test and were found to be non-normally distributed. Therefore, all the continuous data are presented as mean ± standard deviation, and categorical variables are presented as frequency and percentage or median with interquartile range (IQR). The Mann–Whitney U test was used to compare continuous variables with mortality, the chi-square test and cross-tabulation were applied for analyzing the association of categorical variables with mortality across severity groups. The remaining 8 relevant factors (7 laboratory parameters and age) were analysed by Multivariate Logistic Regression. Odds Ratios (OR) with 95% Confidence Intervals (CI) were calculated. A Receiver Operating Characteristic (ROC) curve analysis was also performed to identify the ability of AST, De Ritis ratio, total bilirubin, and IL-6 and albumin levels to predict mortality. The cut-off values were calculated using the the Youden index. The survival probabilities are displayed on a Kaplan–Meier plot and were compared with a log-rank test. A *p*-value < 0.05 was defined as statistically significant.

## 3. Results

### 3.1. Patient Characteristics

A total of 322 laboratory-confirmed COVID-19 patients were enrolled in this study. The median age was 66 (IQR 54–77) years; 178 (55.3%) patients were men. On admission, according to our national protocol based on the WHO guidelines, all patients were categorised into four severity groups. Most of the confirmed COVID-19 cases were moderate (56.2%), with similar frequencies of mild (15.5%) and severe (16.1%) cases. Thirty-six (11.2%) COVID-19 patients died in the hospital.

The clinical features are summarized in Table 1. In total, 57 COVID-19 patients (17.7%) already had gastrointestinal (GI) symptoms on admission. The most typical initial GI symptom was diarrhea (9.6%). Among comorbidities, hypertension and diabetes were the most common being present in 53% (171/322) and 31% (100/322) of the patients, respectively.

Thirty (9.3%) COVID-19 patients reported preexisting liver disease in their medical history; two-thirds of the patients with liver disease had liver cirrhosis. To assess the prognosis in patients with cirrhosis, we applied the modified Child–Pugh classification. On admission, out of the 20 patients with cirrhosis, 9 were classified as CP-A (45%), 7 as CP-B (35%), and 4 as CP-C (20%). One laboratory-confirmed COVID-19 patient with CP-B had hepatic decompensation and died due to the onset of acute-on-chronic liver failure (ACLF). The etiological agents of liver cirrhosis were as follows (Figure 1): alcohol, 40% (8/20); hepatitis C virus, 15% (3/20); nonalcoholic steatohepatitis, 10% (2/20); autoimmune hepatitis, 5% (1/20); hepatitis B virus, 5% (1/20).

### 3.2. Laboratory Findings

The laboratory findings of the recovered patients (*n* = 286) vs. those of the COVID-19 (*n* = 36) patients who died are compared in Table 2. The mean values of AST, total bilirubin, CRP, PCT, and IL-6 were significantly higher in the dead patients compared to the COVID-19 survivors (*p* < 0.05). By contrast, the mean level of albumin was significantly lower in the dead patients (*p* < 0.05). The calculated De Ritis ratio and age also showed a significant difference between the two groups (*p* < 0.05), while no differences in comorbidities or GI symptoms were noted.

### 3.3. De Ritis Ratio as an Independent Predictor for In-Hospital Mortality in COVID-19 Patients

As demonstrated in Table 3, AST, De Ritis ratio, total bilirubin, IL-6 and albumin levels, and age were independently associated with in-hospital mortality. The De Ritis ratio proved to be an independent risk factor for in-hospital mortality with an OR of 29.967 (*p* < 0.001; 95% CI 5.266–170.514).

### 3.4. Predictive Value of the De Ritis Ratio for In-Hospital Mortality

The ROC curves of AST, total bilirubin, IL-6, albumin, and De Ritis ratio are compared in Figure 2. The AUC value of the De Ritis ratio (AUC = 0.850, 95% CI 0.777–0.923, *p* < 0.05) was the highest among those investigated. The optimal cut-off value was 1.218, with a sensitivity of 80.6% and a specificity of 75.2% (Table 4).

We conducted a Kaplan–Meier analysis of survival for COVID-19 patients stratified by the De Ritis ratio (Figure 3). A De Ritis ratio ≥ 1.218 was significantly associated with a shorter survival and thus with a higher in-hospital mortality (log-rank test: *p* < 0.001).

As demonstrated in Table 5, a De Ritis ratio ≥ 1.218 was significantly associated with age, disease severity grade (Figure 4), and increased occurence of ascites. In addition, in patients with a De Ritis ratio ≥ 1.218, we found significantly higher AST and IL-6 levels but significantly lower ALT levels compared to patients with a De Ritis ratio < 1.218.

## 4. Discussion

The identification of prognostic biomarkers for progression in hospitalized COVID-19 patients has been a high priority during the pandemic. Therefore, we conducted a retrospective study to analyse the patient characteristics and laboratory parameters used for the assessment of prognosis in COVID-19 patients. We present a single-center cohort of 322 patients with COVID-19, which is one of the largest analyzed on this matter in the Central European region that we are aware of.

A slight majority of the patients was male, and on admission, more than half of the cases were classified as moderate. Of the 322 cases, 16.1% were severe, which is similar to the figure reported in international studies [20,21]. Thirty-nine critically ill patients were treated in an intensive care unit due to an excessive proinflammatory response with multiorgan failure, known as cytokine release syndrome [22,23].

A recent prospective study of 20,133 UK patients and a cohort study of 192,550 US patients reported hospital mortality rates of 26% and 13.6%, respectively [24,25]. By contrast, the death rate in our study was 11.2%. In previous studies, 11.3–50.5% of COVID-19 patients presented with gastrointestinal symptoms, which is similar to our findings [26,27,28]. Of our patients, 57 (17.7%) presented with a digestive symptom, predominantly with diarrhea. Decreased appetite was excluded from the study, because it is not a well-defined gastrointestinal symptom.

Singh and colleagues reported that patients with preexisting liver disease, especially those with cirrhosis, were more susceptible to a poor outcome compared to patients without liver injury [29]. In our cohort, two-thirds of the patients with liver disease had cirrhosis. Being the largest hepatology center in Hungary, our department routinely provides care for patients with cirrhosis, in particular Child-B and Child-C chirrosis, which may explain the high proportion of patients with cirrhosis in the cohort. The overall prevalence of chronic liver disease in patients with COVID-19 was 9.3%, also in line with previously published numbers [30,31]. Relevant to cirrhosis, patients with alcohol use disorder are particularly vulnerable and might be among the populations most seriously affected by COVID-19 due to their dysfunctional immune system; their status during the pandemic was aggravated by postponed medical checkups and often by alcohol relapse.

Previous studies reported that significantly higher levels of liver transaminases (AST, ALT), total bilirubin, cholestatic liver enzymes (GGT, ALP), and inflammatory markers (CRP, IL-6, PCT), as well as lower albumin levels, were associated with poor outcome [32,33,34]. In accordance with international data, we found significantly elevated AST, total bilirubin, CRP, PCT, IL-6 levels and significantly lower albumin levels in deceased patients following COVID-19 infection (Table 2). However, while ALT was not significantly higher in deceased patients compared to survivors, the AST/ALT ratio was found to be significantly associated with in-hospital mortality. Elevated AST is less specific to liver injury compared to ALT but may indicate multiple organ dysfunction [35]. The De Ritis ratio proved to be a valuable warning indicator of liver damage. In a metaanalysis of 4606 patients, an elevated De Ritis ratio was found to be associated with poor prognosis in COVID-19 patients, with a sensitivity of 55% (95% CI 36–73), a specificity of 71% (95% CI 52–85), and an AUC of 0.67 (95% CI 0.63–0.71) [36]. In our findings, the De Ritis ratio on admission could predict a fatal outcome with above-average sensitivity and specificity. However, the cut-off value in our study is lower than that for previously published international cohorts [37,38]. In a survival analysis, patients with a De Ritis ratio ≥ 1.218 had significantly worse chances to survive and a 2.3-fold higher risk of poor outcome (Figure 5). In addition, compared to patients with a De Ritis ratio < 1.218, patients with a De Ritis ratio ≥ 1.218 had significantly higher AST and IL-6 levels. Elevated IL-6 levels were correlated with excessive immune response, hyperinflammation, and poor prognosis in COVID-19 patients [39]. In a prospective study of immune-inflammatory biomarkers conducted on 153 COVID-19 patients, IL-6 was confirmed as the most accurate and highly predictive inflammatory biomarker of mortality [40]. In our analysis, an elevated De Ritis ratio on admission was independently associated with a more than 29-fold higher chance for in-hospital mortality.

Altogether, our data provide evidence that a De Ritis ratio ≥ 1.218 on admission is a highly sensitive prognostic marker to predict the in-hospital mortality of COVID-19 patients. An elevated De Ritis ratio with high IL-6 levels may indicate the dysregulation of the immune response and the onset of multiple organ failure, including liver injury; therefore, this parameter might serve as a red flag indicating the exacerbation of a COVID-19 infection.

## 5. Limitations

The main limitations of the study include its retrospective single-center design and relatively small sample size. The number of enrolled COVID-19 patients with liver disease, in particular with liver cirrhosis, was limited. Hence, further prospective studies are needed to evaluate the diagnostic utility of an elevated De Ritis ratio in this subgroup. In addition, laboratory parameters including INR, LDH, WBC, CK, direct bilirubin, or hemoglobin were excluded due to the unavailbility of laboratory tests. Therefore, these laboratory markers will also need to be monitored in the whole study population in future prospective studies.

## 6. Conclusions

In conclusion, an elevated De Ritis ratio on admission is independently associated with mortality in COVID-19 patients. Patients with a De Ritis ratio ≥ 1.218 are more susceptible to liver damage and cytokine release syndrome. We also propose that the De Ritis ratio and the IL-6 level should be employed in guidelines for the risk stratification of COVID-19 patients to assist medical providers in decision making.

## Figures and Tables

**Figure 1 viruses-14-02360-f001:**
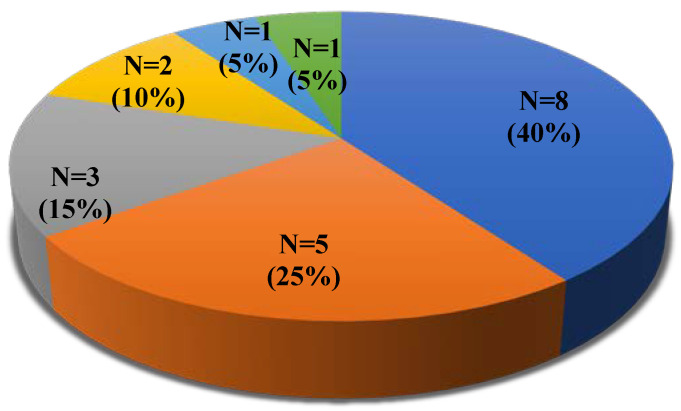
Distribution of etiological agents in COVID-19 patients with liver cirrhosis. HCV, hepatitis C virus; NASH, nonalcoholic steatohepatitis; HBV, hepatitis B virus; AIH, autoimmune hepatitis.

**Figure 2 viruses-14-02360-f002:**
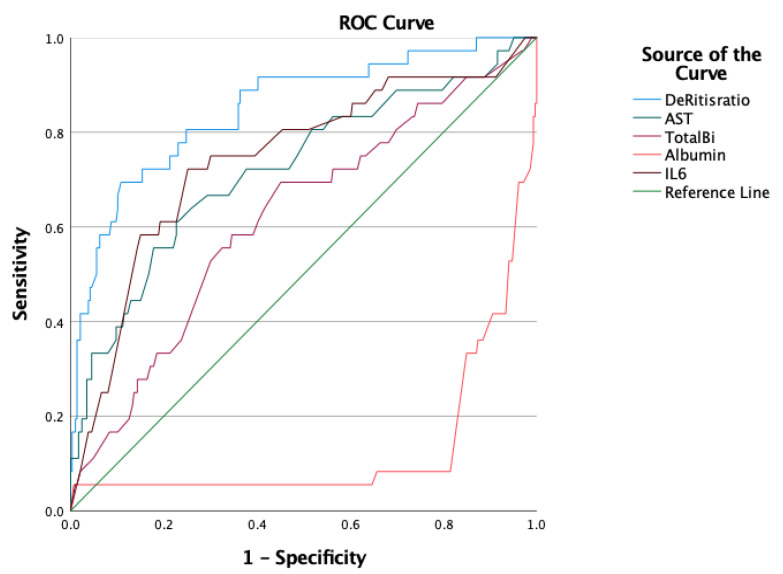
ROC curves of De Ritis ratio, AST, total bilirubin, albumin, and IL-6. ROC, Receiver Operating Characteristic; AST, aspartate aminotransferase; IL-6, interleukin 6.

**Figure 3 viruses-14-02360-f003:**
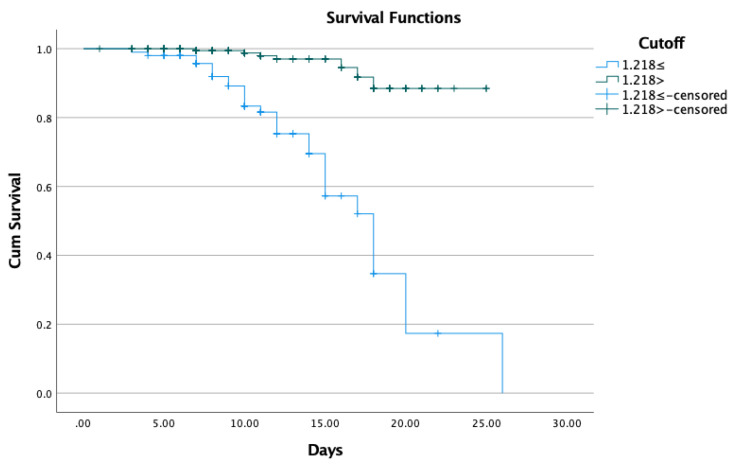
Kaplan–Meier survival analysis of COVID-19 patients stratified by the De Ritis ratio using a cut-off value of 1.218.

**Figure 4 viruses-14-02360-f004:**
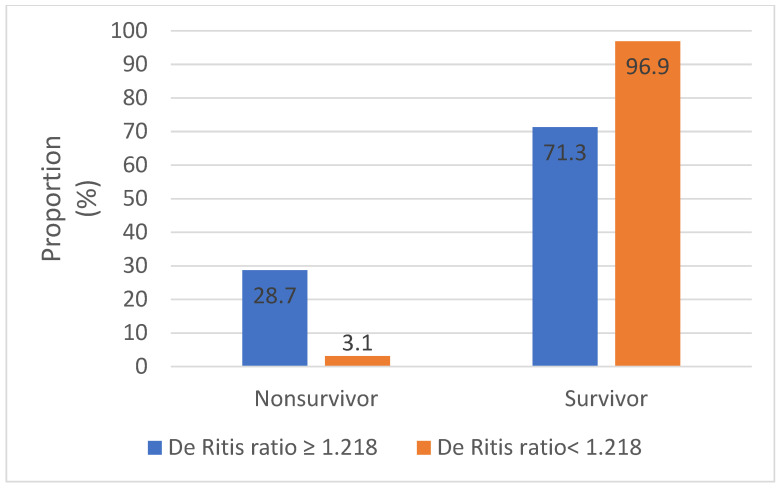
Association between De Ritis ratio and mortality in COVID-19 patients.

**Figure 5 viruses-14-02360-f005:**
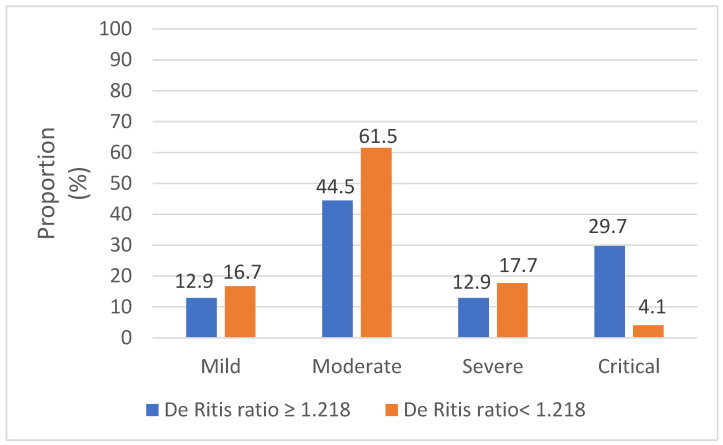
Association between De Ritis ratio and disease severity grade in COVID-19 patients.

**Table 1 viruses-14-02360-t001:** Epidemiology, clinical characteristics, and severity grade of 322 COVID-19 patients admitted to the Semmelweis University Department of Surgery, Transplantation, and Gastroenterology between November 2020 and March 2021.

Epidemiological, Clinical Characteristics	Patients (*n* = 322)
* Age (years)	66 (54–77)
Gender (male/female), *n* * Admission time (days)	178/144 4 (3–6)
Hospital stay (days)	11 (8–14)
GI symptoms *n* (%)	57 (17.7)
Diarrhea, *n* (%) Vomit, *n* (%) Melaena, *n* (%) Ascites, *n* (%)	31 (9.6) 5 (1.6) 5 (1.6) 7 (2.2)
Hipertension, *n* (%)	171 (53)
Diabetes, *n* (%)	100 (31)
Cancer, *n* (%) Anaemia, *n* (%)	21 (6.5) 9 (2.8)
Liver disease, *n* (%) cirrhosis, *n* Child–Pugh A, *n* Child–Pugh B, *n* Child–Pugh C, *n* without cirrhosis, *n*	30 (9.3) 20 9 7 4 10
Severity grade Mild, *n* (%) Moderate, *n* (%) Severe, *n* (%) Critical, *n* (%) In-hospital mortality rate, *n* (%)	50 (15.5) 181 (56.2) 52 (16.1) 39 (12.1) 36(11.2)

* Data shown as median (IQR). IQR, interquartile range.

**Table 2 viruses-14-02360-t002:** Comparison of clinical conditions and laboratory results on admission between COVID-19 recovered patients and COVID-19 patients who died.

Parameter	COVID-19 Recovered Patients *n* = 286	COVID-19 Patients Who Died *n* = 36	*p*
AST, mean (SD)	33.1 (26)	74.2 (78)	**<0.001**
ALT, mean (SD)	37.4 (29.6)	48.4 (75.3)	0.745
De Ritis ratio, mean (SD)	1.0 (.39)	1.8 (.85)	**<0.001**
GGT, mean (SD)	91 (123)	125 (169)	0.437
ALP, mean (SD)	136.9 (126)	189.7 (172)	0.061
Total bilirubin, mean (SD)	33.1 (84.6)	70.8 (154.4)	**0.019**
Albumin, mean (SD)	35.7 (7)	27.9 (13.7)	**<0.001**
CRP, mean (SD)	148.4 (230.9)	260.3 (397.9)	**0.008**
PCT, mean (SD)	6.67 (46.9)	23.1 (92.5)	**<0.001**
IL-6, mean (SD)	39.6 (43.3)	86.4 (60.7)	**<0.001**
Age, mean (SD)	63 (16)	79 (10)	**<0.001**
Admission time, mean (SD) Hospital days, mean (SD)	5 (2.5) 11 (5)	4 (2) 13 (5)	0.495 0.077
Diarrhea, *n* (%)	30 (10.4)	1 (2.7)	0.139
Ascites, *n* (%)	5 (1.7)	2 (5.5)	0.140
Hipertension, *n* (%)	154 (53.8)	17 (47.2)	0.453
Diabetes, *n* (%)	91 (31.8)	9 (25)	0.405
Liver disease, *n* (%)	26 (9)	4 (11.1)	0.694
Cirrhosis, *n* (%)	17 (5.9)	3	0.576

Statistically significant values are presented in bold. AST, aspartate aminotransferase; ALT, alanine aminotransferase; GGT, gamma-glutamyl transferase; ALP, alkaline phosphatase; CRP, C-reactive protein; PCT, procalcitonin; IL-6, interleukin 6; SD, standard deviation.

**Table 3 viruses-14-02360-t003:** Logistic regression for in-hospital mortality comparing seven laboratory parameters and age.

Variable	β	SE	*p*	OR	CI 95%
AST	0.034	0.010	** <0.001 **	1.034	1.015–1.054
De Ritis ratio	3.400	0.887	** <0.001 **	29.967	5.266–170.514
Total bilirubin	0.008	0.003	**0.003**	1.008	1.003–1.013
CRP	−0.001	0.002	0.336	0.999	0.996–1.002
PCT	0.002	0.005	0.613	1.002	0.993–1.012
IL-6	0.027	0.008	** <0.001 **	1.027	1.012–1.042
Albumin	−0.293	0.058	** <0.001 **	0.746	0.666–0.836
Age	0.129	0.032	** <0.001 **	1.138	1.069–1.211

Statistically significant values are presented in bold. AST aspartate aminotransferase; CRP C-reactive protein; PCT procalcitonin; IL-6 interleukin 6; S.E standard error; OR odds ratio; CI confidence interval.

**Table 4 viruses-14-02360-t004:** Diagnostic accuracy of the five investigated laboratory parameters.

Prognostic Marker	AUC (95% CI)	Cut-Off	Sensitivity	Specificity	*p*
AST	0.723 (0.624–0.821)	29.5	0.722	0.622	**<0.05**
De Ritis ratio	0.850 (0.777–0.923)	1.218	0.806	0.752	**<0.05**
Total bilirubin	0.619 (0.519–0.719)	10.1	0.722	0.437	**<0.05**
IL-6	0.743 (0.649–0.837)	51.915	0.722	0.748	**<0.05**
Albumin	0.133 (0.057–0.208)	29.4	0.361	0.126	**<0.05**

Statistically significant values are presented in bold. AST aspartate aminotransferase; IL-6 interleukin 6; AUC area under curve; CI confidence interval.

**Table 5 viruses-14-02360-t005:** Demographics, clinical and laboratory characteristics of COVID-19 patients grouped by the optimal cut-off value of the De Ritis ratio.

Variable	De Ritis Ratio ≥ 1.218 *n* = 101	De Ritis Ratio < 1.218 *n* = 221	*p*
Mortality rate, *n* (%)	29 (28.7)	7 (3.1)	** <0.001 **
Severity grade Mild, *n* (%) Moderate, *n* (%) Severe, *n* (%) Critical, *n* (%)	13 (12.9) 45 (44.5) 13 (12.9) 30 (29.7)	37 (16.7) 136 (61.5) 39 (17.7) 9 (4.1)	** <0.001 **
Gender (male/female), *n*	50/51	128/93	0.159
GI symptoms *n* (%) Diarrhea, *n* (%) Ascites, *n* (%)	19 (18.8) 7 (6.9) 5 (5)	38 (17.2) 24 (10.9) 2 (0.9)	0.724 0.267 **0.021**
Hypertension, *n* (%)	51 (50.5)	120 (54.3)	0.526
Diabetes, *n* (%)	30 (29.7)	70 (31.7)	0.723
Liver disease, *n* (%) cirrhosis, *n*	9 (8.9) 6 (5.9)	21 (9.5) 14 (6.3)	0.865
AST, mean (SD)	45.9 (39)	34 (36.8)	** <0.001 **
ALT, mean (SD)	27.9 (19.8)	43.5 (42.4)	** <0.001 **
Total bilirubin, mean (SD)	41 (98.3)	35.6 (94.1)	0.133
Albumin, mean (SD)	34 (9.6)	35.2 (7.7)	0.088
CRP, mean (SD)	158.5 (256)	162 (257.4)	0.539
PCT, mean (SD)	13.2 (68)	6.3 (46.2)	0.449
IL-6, mean (SD)	56.9 (55.7)	39.4 (42.8)	**0.006**
Age, mean (SD)	70 (16)	63 (16)	** <0.001 **

Statistically significant values are presented in bold. AST, aspartate aminotransferase; ALT, alanine aminotransferase; CRP, C-reactive protein; PCT, procalcitonin; IL-6, interleukin 6; SD, standard deviation; GI, gastrointestinal.

## Data Availability

Accessible upon reasonable request from the corresponding author.

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
