# Peer review of "The Elevated De Ritis Ratio on Admission Is Independently Associated with Mortality in COVID-19 Patients"

_viruses, 2022, doi:10.3390/v14112360_

Round 1

Reviewer 1 Report

The MS presented a study that De Ritis ratio on admission is independently associated with mortality in COVID-19 patients. The conclusion would be helpful to unstand COVID-19 and may be useful for clinics. However, some issues should be figured out before publication.

1.  The admission time should be listed because the levels of  biomarkers are related to the illness duration

2. Did the admission time has any significent difference between COVID-19 recovered patients and COVID-19 dead patients in table 2?

3. How about the corelation between  De Ritis ratio and age in COVID-19 patients?  

Reviewer 2 Report

The paper submitted by Drácz B et al., shows nice data on association of De Ritis ratio on admission with mortality in COVID in patients. These results came to sustain limited literature data suggesting the potential use of De Ritis ratio as biomarker for disease evolution. De Ritis ratio was previously associated with mortality in different pathologies (such as liver disease, cancer, sepsis), that were found to be risk factors in COVID. The study has some limitation and the authors are presenting them in the end of the article, such as sample size, inability to link to subgroup of patients at risk, lack of some lab determinations (Neutrophil to Lymphocytes ratio).

 Minor

1.       Check Figure 4 and 5 for classification of patients in <1.28 and >1.28, it does not match to the data from table 4

2.       The last paragraph in the discussion section should be reformulated, there are data (even if the cut off was different) showing the association of De Ritis ratio with mortality in COVID-19.

“Our data are the first to show that De Ritis ratio ≥1.218 proved to be a highly sensitive prognostic marker to predict in-hospital mortality. The elevated De Ritis ratio with higher IL-6 may indicate both liver injury and dysregulation of immune response and might serve as a red flag that COVID-19 infection may exacerbate. “

The discussion that high De Ritis ratio might reflect liver damage; this is still a matter of debate, since AST might also be released upon injury of other tissues/cells (such as skeletal muscles, heart, immune cells) and ALT was not significantly higher in deceased patients. Please discus you point of view.

Round 2

Reviewer 1 Report

Please make double check on the discription  “Previous studies reported that significantly higher levels of liver transaminases (AST, ALT), total bilirubin, cholestatic liver enzymes (GGT, ALP), and inflammatory markers (CRP, IL-6, PCT), as well as lower albumin, were associated with poor outcome [32-34]. We found significant differences in laboratory parameters including serum AST, total bilirubin, albumin, CRP, PCT, IL-6 in relation to mortality” to be sutable or not. The results do not suport the " difference" on all the mentioned  biomarkers.
